# Exploratory Analysis of Nutritional Quality and Metrics of Snack Consumption among Nepali Children during the Complementary Feeding Period

**DOI:** 10.3390/nu11122962

**Published:** 2019-12-04

**Authors:** Alissa M. Pries, Elaine L. Ferguson, Nisha Sharma, Atul Upadhyay, Suzanne Filteau

**Affiliations:** 1Helen Keller International USA, One Dag Hammarskjold Plaza, Floor 2, New York, NY 10017, USA; 2London School of Hygiene and Tropical Medicine, Keppel St, Bloomsbury, London WC1E 7HT, UK; Elaine.Ferguson@LSHTM.ac.uk (E.L.F.); Suzanne.Filteau@lshtm.ac.uk (S.F.); 3Helen Keller International Nepal, Green Block, Ward No 10, Chakupat, Patandhoka Road, Lalitpur 44700, Nepal; NSharma@hki.org (N.S.); AUpadhyay@hki.org (A.U.)

**Keywords:** snack, dietary measurement, infant and young child feeding, Nepal

## Abstract

The World Health Organization recommends feeding snacks between meals to young children. This study explored nutritional quality of snacks consumed between meals and consumption metrics (% total energy intakes (%TEI) and amount of kcal from snacks) to understand correlations with dietary outcomes (total energy intakes and dietary adequacy) and body-mass-index-for-age z-scores (BMIZ). Data used were 24-h dietary recalls and anthropometric measurements among a representative sample (*n* = 679) of one-year-olds in Nepal. Nepali meal patterns for young children were identified through formative research and all foods/beverages consumed outside of meals were categorized as snacks. A nutrient profiling model was used to categorize snacks as healthy or unhealthy, based on positive and negative nutrient content. Snacks consumed between meals provided half of all energy consumed, and were associated with increased energy and nutrient intakes. The positive effect of snacks between meals on dietary adequacy was greater when these snacks were healthy, while increasing %TEI from unhealthy snacks consumed between meals was negatively associated with dietary adequacy. Consumption of snacks between meals was not associated with mean BMIZ among the children. These findings indicate that the provision of and nutritional quality of snacks are important considerations to communicate to caregivers. Discouragement of unhealthy, nutrient-poor snacks is critical for complementary feeding dietary guidelines in contexts experiencing nutrition transition.

## 1. Introduction

As food environments evolve globally, many populations are experiencing a nutrition transition whereby diets are increasingly characterized by snacking [1,2]. Consumption of unhealthy snack foods among children has been noted across both high-income [3,4,5,6,7,8,9,10] and low- and middle-income countries [11,12,13,14]. While snacks, in addition to meals, can be an important source of energy and nutrients, consumption of ultra-processed snack foods and beverages is concerning and the role of snacks in the escalating rates of overweight and obesity among children has garnered substantial attention [15,16]. In emerging economies, the growing availability of unhealthy foods has been attributed to shifting dietary patterns and displaced consumption of other more nutritious foods [17]. In such contexts, unhealthy snacks could contribute to not only overnutrition, but increased risk of undernutrition and reduced dietary adequacy.

The significance of how ‘snacks’ are defined must be carefully considered, both in research related to the effect of snack consumption on nutrition and also, in dietary recommendations for snacks. Furthermore, the contextual relevance of definitions within global populations that have varying food cultures must also be considered. The definitions of snacks used in research and dietary guidelines are wide-ranging [18,19]. Recent reviews found that common definitions of ‘snacks’ were based on time of consumption or nutritional quality of foods consumed, with little consensus across the literature [19,20]. Even within these definitions, there is wide variation in how they are operationalized. Studies regarding snacks consumed outside of meals vary on how meal patterns are defined—some opting for participant designation of meals versus snacks [21,22] and others using predetermined timetables [1,23,24]. Those that assess nutritional quality of snacks also vary on whether foods and/or beverages are included in the definition [3,12,25] and how nutritional quality of these foods/beverages is considered, with some evaluating nutrient content and others assuming low nutritional quality based on food types (i.e., junk foods or sugar-sweetened beverages). Without adequate and consistent consideration of nutritional quality, subsequent dietary recommendations around snacks may be incongruent. 

The wide range of metrics for quantitative description of snack consumption is also problematic and merits consideration. In a systematic review assessing snack food/beverage consumption among young children [26], the metrics used to estimate snack consumption across studies ranged from the number of snacks consumed per week or per day to the contribution of snack foods to total energy intake (%TEI). This variation of definitions for snacks and metrics used to measure consumption makes it challenging to draw conclusions about the role of these foods in diets and their nutritional implications. In their review exploring how snack definitions affect the ability to interpret literature, Johnson and Anderson [18] identified 26 studies assessing the relationship between snacking and obesity whose varied snacks definitions precluded any comparison of findings across studies. Analysis by Gregori et al. [27] found that varying snacks definitions and metrics of consumption resulted in vast variability (70%) in results on associated probability of obesity among children and adolescents. The variability in results stemming from varying snack definitions and consumption metrics across studies assessing other dietary and nutritional outcomes, such as dietary inadequacy and linear growth, is unknown. It has been noted that research which specifically investigates the effect of definition/metrics variation on study findings is extremely limited [19].

Additionally, in childhood, the role of snacks in dietary adequacy and consequent relationships with nutritional status may vary across age groups and context. Much research has been focused on the relationship between snacks and weight status among primary-school-age children [28,29,30] and adolescents [23,31,32]. Only limited research has been done on the contribution of snack consumption to dietary adequacy and nutritional outcomes among children under two years of age, whose energy and nutrient requirements are high and gastric capacity limited [33]. This lack of evidence also likely contributes to the void of guidance on snacks for young children during the complementary feeding period. While dietary guidelines specific to this young age group have been recently developed [34,35], they are generally limited in number [36] and few provide specific guidance around healthy versus unhealthy snack foods and beverages. Furthermore, in low- and middle-income country (LMIC) contexts, access to nutrient-dense foods can be limited and so the effect of nutritional quality of snacks consumed between meals may be different for these children than those in high-income settings where overnutrition is the main concern. 

Undernutrition among children in Nepal remains high. Nationally, 36% and 10% of children under 5 years of age are stunted and wasted, respectively [37]. However, Nepal is in the beginning stages of a nutrition transition as overweight/obesity among adults is rising—increasing from 9% to 22% among women of reproductive age between 2006 and 2016 [37]. Consumption of commercially packaged snack foods is prevalent among young children in Nepal. In a 2014 Kathmandu Valley study, 57% and 43% of children 6–23 months of age were consuming commercially produced biscuits/cookies and sweets/candy, respectively [14]. Consumption of these commercially produced snack foods was higher than consumption of dark green leafy vegetables (35%), orange-fleshed fruits (1%) and vegetables (8%), and eggs (24%).

The aim of this paper was to examine how nutritional quality of snacks consumed between meals relates to overall dietary adequacy/total energy intakes (primary objectives) and body mass index-for-age (secondary objective). In addition, this paper explored how metrics of snack consumption modify conclusions regarding these relationships. The metrics of snack consumption examined were (a) %TEI from snacks and (b) the amount of energy from snacks. These analyses were done using data collected from one-year-old children living in Kathmandu Valley, Nepal. This exploratory analysis was conducted to highlight the importance of understanding nutritionally relevant snack parameters and appropriate metrics with the aim of informing dietary guidelines and programmatic messaging around snacks for young children in a low- and middle-income country context experiencing nutritional transition.

## 2. Materials and Methods

### 2.1. Study Design and Sampling

A cross-sectional quantitative survey was conducted from February to April 2017 among a representative sample of Nepali primary caregivers and their 12–23-month-old children. Participants (*n* = 745 dyads) were selected, using multi-stage cluster sampling, from municipality wards across Kathmandu Valley. The sampling strategy and sample size calculations are detailed elsewhere [38]. In summary, 78 clusters of an anticipated 10 caregiver–child pairs were assigned across wards based on probability proportional to size (PPS). For inclusion in the study, children had to be 12–23 months of age on the day of interview, children and their primary caregivers had to be current residents of Kathmandu Valley. Caregiver–child pairs were excluded if the child was severely ill or if they had a congenital/physical malformation that inhibited feeding.

Participants within each cluster were recruited 2–3 days prior to the scheduled day of data collection by a trained recruitment team using standardized methods. On the day of interview, interviewer-administered questionnaires and a 4-pass interactive 24-h dietary recall (24HR) were conducted in the caregivers’ home in order to ensure a comfortable environment and to also aid portion size estimation by using household utensils. Interviews lasted approximately 90 min, after which the caregiver and child would be guided to a central location in the municipality ward for anthropometric measurements. 

The study was conducted in accordance with the Declaration of Helsinki, and the protocol was approved by the London School of Hygiene and Tropical Medicine (reference 11719) and the Nepal Health Research Council (reference 563). Written informed consent was obtained from all caregivers prior to participation in the study.

### 2.2. Data Collection and Methods

Through interviewer-administered questionnaires, data were collected on demographic and socio-economic characteristics pertaining to the caregiver and child. The questionnaire was translated into Nepali, back-translated into English, and pre-tested prior to data collection. An interactive, quantitative, 4-pass 24HR interview was conducted with each caregiver to gather dietary intake data for their child 12–23 months of age during the day prior to interview [39]. Recalls were collected for every child, and a repeated measure was collected on a non-consecutive day among a random sample of 10% of the children to account for intra-individual variation. Recalls were conducted on all days of the week to account for the day-of-the-week effect, and repeated recalls were typically conducted 2–3 days after the initial dietary recall. Dietary data collection was conducted within one agricultural season (early February to early April 2017) in order to minimize variation in diets across the data collection period. Full details of questionnaire design and dietary assessment methods are presented elsewhere [38]. Recumbent length and weight of children were measured by two trained nurses using standardized procedures [40] with length/height boards (Shorr boards) and SECA scales (model 878 U; ± 0·1kg precision). Two serial measures for recumbent length and weight were taken. If the two measures of length differed by more than 0.5 cm or if weight measures differ by more than 0.5 kg, these measurements were discarded, and the two serial measurements were taken again. The measurements were entered and averaged for analysis. 

### 2.3. Parameters of Snacks

The parameters of the snacks explored were first based on consumption between meals and subsequently, a sub-category analysis of nutritional quality of these snacks. These two parameters were chosen because guidelines for infant and young children feeding reference both aspects in relation to optimal diets [41,42]. 

Consumption between meals: During the complementary feeding period, the WHO defines a ‘snack’ as a food/beverage consumed between meals [43]. This parameter is a common definition for ‘snacks’ among all age groups in scientific literature [20]. During the complementary feeding period, the recommendation to feed both meals and snacks acknowledges the importance of frequent, small feeds to accommodate limited gastric capacity of young children [44]; feeding of snacks is an opportunity for additional energy and nutrients during a vital period of growth and development. 

In our study, to define snacks as foods/beverages consumed between meals, the typical meal pattern for young Nepali children was first established through formative research involving structured observations of ten children 12–23 months of age living in Kathmandu Valley. For structured observations, five neighborhoods across different municipality wards were selected, and two households with a child 12–23 months of age were purposively sampled from each neighborhood. Written informed consent was obtained from all caregivers who agreed to participate. Observations were conducted over a 12-h period with two research assistants assigned to a household. To account for day-of-the-week effect, observations were conducted on all days of the week. The child was followed throughout the day, and all activities noted. In contrast to the typical meal pattern among Nepali adults, which is based on two main meals of the day (referred to as *khana* and consisting of rice, dal, and seasonal vegetables), children were typically fed multiple times throughout the day and received 3 meals with small feeds in-between. Children were fed not only the two meals of morning and evening *khana*, but were also fed an afternoon meal. The findings from these structured observations regarding meal patterns for young children were further explored during seven focus group discussions among Nepali caregivers (grandmothers and mothers) of children 12–23 months of age, during which caregivers confirmed this pattern of feeding. Using this observed meal pattern, foods and beverages not consumed during meals were categorized as snacks, meaning they were consumed not during *khana* meals and not during the afternoon meal. 

Nutritional quality of snacks: Use of food categories is a common approach to assess nutritional quality, whereby a taxonomy of types of foods and beverages is used to group them based on the assumed nutritional-quality of these foods [27]. The WHO has highlighted the increasing presence of unhealthy snack foods/beverages in young children’s diets and noted the need for an indicator of consumption of such foods as a measure of unhealthy diets [42]. Distinguishing nutritional quality based solely on food categories can be problematic as it relies upon many assumptions [45], but the use of nutrient profiling has aided this definition by providing a more structured evaluation of the nutritional composition of foods/beverages. 

To assess nutritional quality of snacks, foods and beverages consumed between meals (based on the meal pattern timings noted above) were identified as snacks, and these individual snack items were then sub-categorized as ‘unhealthy’ or ‘healthy’ using a nutrient profiling model from the United Kingdom’s Food Standards Agency (UK FSA) [46]. The UK FSA model evaluates the presence and degree of both ‘negative’ nutrients (energy, total sugar, saturated fat, and sodium per 100 g of food/beverage) and ‘positive’ nutrients (fiber and protein per 100 g of food/beverage, and % of the food/beverage that is fruit/vegetable/nut) and uses an algorithm to determine a score for each food, and a cut-off is used to categorize foods as unhealthy or healthy based on this score.

In theory, consumption of unhealthy snack foods and beverages between meals provides energy-density but not nutrient density and may displace micronutrient intake. Therefore, the nutritional quality of snacks consumed between meals was explored to assess if any positive associations with dietary/nutritional outcomes from the WHO recommended temporal definition of snacks are moderated by the consumption of unhealthy versus healthy snacks.

### 2.4. Snack Consumption Measurement

Two metrics of snack consumption were used for our analysis: (1) amount of energy intake from snacks (snack kcal) and (2) proportion of total energy intake from snacks (snack % TEI). We hypothesized that total energy intakes from snacks would be a better predictor of overnutrition-related outcomes, while % TEI from snacks would be a better predictor of micronutrient displacement and dietary adequacy. Total energy intakes from non-breastmilk foods were calculated based on intake of all foods and beverages reported by caregivers during the 24HR. Because breastmilk consumption was quantified in the survey, the outcome of interest (total energy intakes) will differ between breastfed and non-breastfed children; therefore, children who were not breastfed were excluded from this analysis. 

### 2.5. Diet and Nutritional Status Outcome Variables

Dietary outcomes of interest included: (1) total energy intakes from non-breastmilk foods and (2) mean probability of dietary adequacy (MPA) for 11 micronutrients based on 24HR measured intakes of non-breastmilk foods and estimated intakes of breastmilk. Energy and nutrient intakes were calculated using 24HR data and a compiled food composition table (FCT). Breastmilk intake was calculated at the group level, and this amount (293 g/d) was added to the diet of each breastfeeding child. It was calculated by subtracting the median energy intake from non-breastmilk foods from their estimated average energy requirements, using their average body weight of 9.7 kg, assuming the energy content of breastmilk was 70 kcal/100 g [(800 kcal–595 kcal)/70 kcal/100g × 100 = 293 g]. The energy and nutrients contributed by 293 g of breastmilk were added to the energy and nutrient intakes contributed by non-breastmilk foods for all breastfed children. 

The MPA across the 11 micronutrients was calculated for each child. After inter- and intra-individual variance was calculated based on the repeated 24HR, best linear unbiased predictor (BLUP) of usual intakes were generated by PC-SIDE for each micronutrient to calculate probability of adequacy (PA) for each child using the Institute of Medicine (IOM) probability approach [47]. IOM Estimated Average Requirements (EAR) were used for calcium, thiamin, riboflavin, niacin, folate, and vitamins A, C B6, and B12 [48]. The International Zinc Nutrition Consultative Group (iZiNCG) EAR was used for zinc, assuming low bioavailability [49]. The iron requirements for young children are not normally distributed, so the probability approach (PA) was calculated based on probabilities of adequacy for intake intervals [50], assuming 5% bioavailability. Body-mass-index z-scores (BMIZ) were calculated using the WHO growth standards [51]. 

### 2.6. Data Analysis

Stata 15 was used to analyze all data. Proportions and means ± (SD) were calculated to describe the sample and snack consumption patterns, and medians with interquartile ranges (IQR) calculated for non-normally distributed data. Non-normally distributed outcome data were log transformed prior to analysis. The relationships between consumption of snacks and diet/nutritional status outcomes were assessed using linear regression with random effects to account for cluster sampling. All models included a set of covariates identified a priori: child sex, child age, wealth status, caste, and caregiver educational attainment. Variance inflation factors (VIF) were used to explore collinearity of models.

Because repeated 24HRs were not conducted across the entire sample, analysis at an individual level was not appropriate and so a population-level analysis (tertiles of snack consumption) was used. For both total kcal from snacks and %TEI from snacks, tertiles of consumption—low/moderate/high consumption—were created across the entire sample for snacks defined temporally and snacks defined by food type. In each model, the dependent variables (diet and nutritional status outcomes) were regressed against the independent variable of snack consumption tertile and predetermined covariates. The analyses were powered to detect 0.3 SD differences in energy and nutrient intakes (powered at 90%) and a 0.5 SD difference in BMIZ (powered at 80%) between the lowest and highest snack consumption tertiles. We hypothesized that the nutritional quality of snacks and metric used to measure snack consumption would influence the strength and direction of associations with nutritional outcomes, as summarized in Table 1. It was hypothesized that snacks would contribute to greater energy intakes, dietary adequacy (MPA), and BMIZ, as greater consumption of foods between meals would theoretically provide additional energy and nutrients and reflect positively on growth. While unhealthy snack consumption between meals was hypothesized to be negatively associated with MPA and positively associated with energy intakes and BMIZ, as these foods are energy-dense and nutrient-poor and so would increase energy but not micronutrient intakes.

## 3. Results

A total of 745 children 12–23 months living in Kathmandu Valley were included in the primary study; details of recruitment, exclusion, and refusal rates can be found elsewhere [38]. Of these 745 children, 91.1% were breastfed at the time of interview and, therefore, 679 children were included in this analysis. Anthropometric measurements were collected from 98% of these 679 children. Just under half of these children were female and their average age was 17.5 months (Table 2). Mothers were the most common type of primary caregiver, and the majority of caregiver/child households were Hindu (83.1%, *n* = 564) and had a median of four household members. Mean probability of adequacy for 11 micronutrients was 57.0% across all children. Almost one-fifth of children were stunted, while less than one percent were overweight/obese. 

Among all children, nearly one-half of TEI came from snacks consumed between meals (Table 3). Overall, median energy intake from non-breastmilk foods among all children was 595 kcal with a median of 275 kcal consumed as snacks between meals. There was substantial variation of both %TEI from snacks and total energy (kcal) consumed as snacks across tertiles of consumption. Of the median 275 kcal consumed as snacks between meals, approximately two-thirds were from foods/beverages categorized as healthy based on their nutrient profiles. These healthy foods/beverages included: fruits, eggs, grains and legumes (typically eaten in the form of porridges), nuts, and buffalo/cow milk. Snacks that were nutrient-profiled as unhealthy provided approximately one-third of calories consumed between meals, and included the following foods/beverages: commercially produced biscuits, bakery items, instant noodles, cheeseballs, chips, candy, chocolates, juice drinks, sweet teas, and soft drinks. Across a full day, the majority of unhealthy foods and beverages were consumed between meals rather than during mealtime; approximately three-quarters of all kcal consumed from unhealthy foods/beverages in a day were consumed as snacks between meals. 

Models exploring the associations between snack nutritional quality and diet/BMIZ outcomes are presented in Table 4, with comparisons made between the highest and lowest consumption tertiles. When snack consumption was measured based on kcal consumed, snacks consumed between meals were positively associated with total energy intakes, with this positive relationship observed in disaggregated analysis of both healthy and unhealthy snacks. In the adjusted models, total energy intake among the highest-tertile consumers of snacks between meals was 161% greater than the total energy intakes of the lowest-tertile consumers (β: 0.96 [CI: 0.88–1.04]; *p* < 0.001). Consumption of all snacks between meals was also found to be positively associated with MPA, both when consumption was measured as snack %TEI and snack kcal. However, the effect size was substantially greater for consumers with the greatest amount of kcal from snacks between meals as compared to those with the greatest %TEI from snacks consumed between meals (β: 0.32 [CI: 0. 0.29–0.35] versus β: 0.04 [CI: 0.005–0.08]). 

The nutritional quality of snacks consumed between meals moderated both the effect size and directionality of the relationship between snack consumption and dietary adequacy, and also depended on the metric of consumption used. When considering the amount of kcal consumed from snacks, the effect size of the positive relationship between kcal from snacks and greater MPA observed among high consumers was substantially lower when considering unhealthy snacks consumed between meals (8% higher mean MPA among high versus low snack consumers) as compared to healthy snacks consumed between meals (33% higher mean MPA among high versus low snack consumers). When considering %TEI from snacks, mean MPA of children with the highest %TEI from healthy snacks consumed between meals was 15 percentage points higher than children with the lowest %TEI from healthy snacks, (*p* < 0.001), while mean MPA of children with the highest %TEI from unhealthy snack foods and beverages was 9 percentage points lower than the mean MPA of children with the lowest %TEI from unhealthy snacks (*p* < 0.001). 

High consumption of snacks between meals, regardless of whether they were nutrient profiled as healthy or unhealthy, was not associated with mean BMIZ. 

## 4. Discussion

Our findings indicate that the consumption of snacks (food/beverages between meals) is associated with higher energy intakes among young children in an urban Nepal setting; however, the nutritional quality of foods/beverages consumed as snacks moderates the relationship between consumption of snacks and overall dietary adequacy in this context where undernutrition is high. In our analysis of Kathmandu Valley breastfed 12–23-month-old children, increasing caloric intakes of and % TEI from all snacks consumed between meals were positively associated with MPA, but the effect on MPA was even greater when these snacks were healthy based on their nutrient profiles. Increasing contribution of unhealthy snack foods and beverages to total energy intakes was negatively associated with dietary adequacy and increasing caloric intakes of these unhealthy foods/beverages between meals substantially reduced the positive effect size of snacks on MPA. Consumption of snacks between meals, healthy or unhealthy, was not associated with mean BMIZ among the children. These findings indicate that the nutritional quality of snacks is an important consideration to communicate to caregivers, as consumption of healthy snacks is associated with greater dietary adequacy, while consumption of unhealthy snacks is associated with a substantially lower dietary benefit. This is critical in a context like Nepal that is experiencing a nutritional transition, and where the nutrient density of snacks is paramount to achieving nutrient requirements for a young child population where stunting is highly prevalent. Updated complementary feeding dietary guidelines that reflect dietary transitions occurring globally are needed, and should consider including specific recommendations to not only encourage healthy snacks, but discourage unhealthy ones. 

Nearly half of all calories consumed from non-breastmilk foods came from snacks consumed between meals, and consuming snacks between meals correlated with greater overall energy intakes, indicating that snacks are an important component of diets among young children in this study. One of the World Health Organization’s guiding principles for complementary feeding is the provision of snacks several times a day between meals to aid achieving energy requirements for young children with limited gastric capacity [44]. In line with this recommendation, one of the key WHO indicators of optimal complementary feeding practices is minimum meal frequency, which includes the count of both meals and snacks fed to a young child [41]. Surveys of dietary energy and nutrient intakes among young children under two years of age, particularly those which also detail snack versus meal eating occasions, are limited [52] challenging the ability to compare to other studies in low-income setting. However, Skinner et al. also note the important role that snacks, in addition to meals, play in young children’s diets. They found that snack-eating occasions (between meals) provided one-quarter of the total energy intake (from breastmilk and all other foods) among US children 1–2 years of age [53]. 

Snacks consumed between meals contributed to greater energy intakes among young children in this study; however, median energy intakes across consumption tertiles were not in excess of energy requirements from non-breastmilk foods given age and average weight. In line with this, none of the snack parameters explored were associated with BMIZ and prevalence of overweight/obesity was extremely low. Mean BMIZ was −0.3, indicating that this sample of Kathmandu Valley children were slightly thinner than would be expected, which could be driven by slightly low energy intakes. In this context, rather than being a risk factor for overnutrition, snacks between meals served as an important opportunity to provide needed calories for children and energy intakes would have likely been inadequate without these additional feeds. Research among older children assessing the link between the snacks consumed between meals and weight status or BMI is consistent with our findings in Nepal. In a study of Malaysian adolescents, Boon et al. reported that consumption of snacks between meals, including fruits, sweets, milk, and soft drinks, had no association with BMI [23] and Gonzalez-Suarez et al. found no relationship between calories from snacks consumed between meals or %TEI from snacks between meals and overweight/obese status among Filipino 10-12-year-olds [54]. However, this latter study did find a relationship between consumption of ‘low-quality snack foods’ (defined as high-fat and/or low-nutrient-density) and ‘sweetened drinks’ and overweight/obesity, indicating that the relationship between snacks and overnutrition is more related to the types of foods consumed. 

The nutritional quality of snacks consumed between meals by young children in this study was highly associated with overall diet quality. Consumption of all snacks was positively associated with MPA, but this effect size was reduced by nearly 75% when snacks between meals were primarily unhealthy foods or beverages. In the adjusted models, diets with greater %TEI from unhealthy snack foods and beverages between meals were associated with lower dietary adequacy. Unhealthy snack food and beverage consumption among children in this study was prevalent, making up one-third of calories consumed between meals. Substantial consumption of micronutrient-poor foods and beverages high in sugar, sodium has been increasingly noted among young children in low-income settings [33] where the nutrient density of complementary foods are typically lower. [55] Separate findings from this Kathmandu Valley study found that the highest consumers of unhealthy snack foods and beverages (regardless of time of consumption) had lower length-for-age z-scores, with this relationship partially mediated by reduced dietary adequacy among the high-snackers [38]. Caregivers of young children are encouraged to feed frequently and read children’s cues for hunger [56]. Often caregivers become concerned with keeping something in the belly and staving off hunger, and may turn to foods that children readily eat [57,58]—but if these foods are nutrient-poor, there may be a detrimental effect on overall diet quality. Results from this study indicate that snacks between meals are a substantial part of young children’s diets and provided nearly half of all calories consumed in a day, but the degree of energy and micronutrients these snacks provide is highly dependent on the nutritional quality of foods/beverages. There is a need to focus infant and young child feeding messaging on snack food quality in addition to frequency of feeds for young children in order to ensure not only adequate energy intakes but also adequate micronutrient intakes. The healthy snacks driving higher dietary adequacy for children in this study specifically included items such as milk, eggs and fruit, which could be highlighted in food-based recommendations for this population. 

Findings from this study indicate that guidance on snacks will be an important consideration as dietary guidelines are developed specifically for young children below two years of age. Though currently limited [36], national food-based dietary guidelines with specific recommendations for infants and young children are increasing and serve as an opportunity to aid caregivers in their snack-feeding practices. The complementary feeding period stretches from 6 months until 2 years of age, and is a period marked not only by the introduction of solid foods but the transition from ‘baby’ foods to ‘family’ foods [52]; food-based dietary guidelines could provide caregivers additional guidance on the types of family foods that are appropriate or inappropriate for young children. The WHO guiding principles for complementary feeding further defines snacks as foods that are ‘usually self-fed, convenient, and easy to prepare’. The characteristics of ‘convenient’ and ‘easy to prepare/feed’ have become hallmark features of commercial snack food and beverage products [59] and are often cited by caregivers as reasons for their use in child feeding [57,60,61]. Additionally, such unhealthy foods are typically lower-cost than other healthier options in most parts of the world [62], and so unhealthy foods as snacks may be financially appealing for many families of young children. It is, therefore, critical for future dietary guidelines to not only continue to recommend snacks for young children, but to increasingly draw attention to the importance of choosing nutritious snacks that are low in added salt/sugar and high in nutrient density. Governments in both New Zealand and Australia have recently published dietary guidelines for infants and toddlers, and in both publications, snacks fed between meals are recommended alongside the recommendation to choose snacks containing limited salt and added sugar [35,63]. A recent global review of 90 food-based dietary guidelines found that while general guidance to limit salt and sugar intakes were nearly universal, only 28% of all national guidelines included messages to limit intakes of processed foods, such as unhealthy snack foods or beverage products [36]. There is a clear need for countries to expand food-based dietary guidelines for the complementary feeding period and to include guidance that can moderate consumption of nutrient-poor, unhealthy snack foods and beverages among young children. 

Results from this paper also highlight the importance of selecting an appropriate metric of snack consumption when exploring relationships with diet and nutritional outcomes. As shown in this Kathmandu Valley study, for research questions exploring whether snack consumption is positively associated with energy intakes the amount of kcal from snacks is an appropriate metric, whereas consumption of snacks based on %TEI would be a more appropriate predictor of dietary adequacy when the nutritional quality of snacks is also considered. A measure of %TEI better represents the overall make-up of a diet and theoretically serves as a better proxy of micronutrient displacement than sole measurement of calories from unhealthy snacks. Other studies among young children have also used this measure when exploring the relationship with dietary intakes [64,65,66]. Furthermore, consumption of snacks based on amount of kcal consumed would be an appropriate metric when exploring the relationship with BMI, while use of %TEI would be less appropriate because there is not necessarily a correlation between %TEI from snacks and overall higher energy intakes that would lead to weight increases. A paper by Gonzalez-Suarez et al. exemplifies the inconsistent relationships between snacks and weight status when not only different definitions but also differing consumption measurements are used [54]. In their paper exploring the association between diets and BMI among Filipino adolescents, %TEI from snacks consumed between meals was not associated with overweight/obesity, while the number of servings of high fat/nutrient-poor snack foods or sweetened drinks was positively associated. Similarly, in a study among Australian 2–18-year-olds, Bell et al. found that %TEI from ‘noncore foods’, such as cakes, biscuits, soft drinks, was not associated with overweight/obesity across age groups but that energy intakes from these foods were positively associated among 2–4-year-olds and similar trends were observed among older children [3]. As researchers continue to explore the relationship between snacks and dietary outcomes, it is crucial that not only snack definition parameters be considered but also that appropriate metrics of consumption be used to contribute to the evidence base for this topic and allow meaningful comparisons across studies.

This analysis has several limitations. First, dietary recalls are subject to error, stemming from potential bias in caregivers’ ability to recall all the foods/beverages consumed by a child. In an attempt to minimize recall error, a pictorial recall-aid was used; however, the ability of such a tool to minimize omissions/additions in a recall should be validated in future research. Additionally, breastmilk intake for children was estimated, and actual intakes were not measured. The use of estimated quantities of breastmilk intake is common in dietary assessments of young children [25,67,68], because of the costs and increased participant burden associated with actual breastmilk intake measurement, but this estimation does introduce error. While some studies estimate breastmilk intake at the individual level, a population-level approach was chosen so that any error associated with this estimation was distributed equally across the sample. With only one 24HR per child, actual energy intake at the individual level could not be established and so estimation of breastmilk intake using a population-level median energy intake was more appropriate for these data. Finally, the cross-sectional design of this study limits any ability to draw conclusions regarding causal relationships.

Despite these limitations, strengths included collection of individual recipe data to allow measurement of differing nutrient content across households, and the development/use of the pictorial recall-aid to reduce omissions. In addition to these considerations for dietary methods, there are also strengths related to sampling and study design. The relatively high response rate (81%) for this survey and randomized sampling procedure potentially minimized selection bias in this study. In addition, the large sample size meant the study was adequately powered to assess the hypothesized relationships between the primary exposure variable and outcomes of interest. 

## 5. Conclusions

These findings indicate that for young children in this urban Nepal setting, what is consumed as a snack between meals is crucial for dietary outcomes. Recommendations for young children to receive frequent feeds with snacks between meals can contribute positively to energy and nutrient intakes, but the benefit of these recommended feeding practices is substantially reduced if the snacks fed are nutrient-poor. Dietary guidelines and young child feeding recommendations should ensure that nutritional quality of snacks and specific discouragement of inappropriate snacks are included in young child feeding recommendations. Future research around the role of snacks in diets and nutritional status should also ensure that the metrics of snack consumption fit research questions and aid meaningful analysis that can provide better understanding globally. 

## Figures and Tables

**Table 1 nutrients-11-02962-t001:** Association testing and hypothesized directionality for models ^1,2^.

		Total Energy Intake	MPA	BMIZ
Model 1:	All snacks consumed between meals + Metric: %TEI	NA	+	NA
Model 2:	All snacks consumed between meals + Metric: kcal	+	+	+
Model 3:	Healthy snacks consumed between meals + Metric: %TEI	NA	+	NA
Model 4:	Healthy snacks consumed between meals + Metric: kcal	+	+	+
Model 5:	Unhealthy snacks consumed between meals + Metric: %TEI	NA	−	NA
Model 6:	Unhealthy snacks consumed between meals + Metric: kcal	+	−	+

^1^ MPA = mean probability of adequacy; BMIZ = body mass for age-index z-score; %TEI = contribution to total energy intake; NA = no association; ^2^ Snacks were defined as foods consumed outside mealtime and nutrient profiling was conducted to categorize these snacks as unhealthy or healthy. Two measurements of consumption were used: (1) total energy (kcal) from snacks and (2) contribution to total energy intake (%TEI) from snacks.

**Table 2 nutrients-11-02962-t002:** Description of primary caregivers and breastfed children 12–23 months of age in Kathmandu Valley, Nepal (*n* = 679) ^1^.

**Caregiver Relationship to Child**	
Mother	92.3 (627)
Grandmother	5.3 (36)
Aunt	1.5 (10)
Other	0.9 (6)
**Caregiver Ethnic Group ^2^**	
Upper caste	39.9 (271)
Advantaged janajati	25.8 (175)
Disadvantaged janajati	27.1 (184)
Dalit/non-dalit terai caste	7.2 (49)
**Caregiver Education**	
No formal education	11.6 (79)
Primary	20.5 (139)
Secondary	52.9 (359)
Tertiary	15.0 (102)
Child age, months	17.5 ± 3.3
Child sex (female)	47.0 (319)
Child total energy intake from non-breastmilk foods (kcal)	595 (415–805)
Child mean probability of adequacy (MPA)	57.0 ± 0.8
Child body-mass-index z-score ^3^	−0.30 ± 0.99
Child overweight/obesity (BMIZ > 2) ^3^	0.9 (6)

^1^ Values presented as % (*n*), median (interquartile range), and mean ± standard deviation; ^2^ upper castes: e.g., Brahman/Chhetri; advantaged janajatis: e.g., Newar, Gurung); disadvantaged janajatis: e.g., Magar/Tamang; dalit/disadvantaged non-dalit terai: e.g., Thakur/Yadav.; ^3^
*n* = 667 for anthropometric measurements.

**Table 3 nutrients-11-02962-t003:** Descriptives of unhealthy versus healthy snacks and consumption measurement, by tertiles of consumption.

	Measurement of Consumption							
	Snack %TEI ^1^				Snack Kcal ^2^			
	All	Low	Mod	High	All	Low	Mod	High
	(*n* = 679)	(*n* = 227)	(*n* = 226)	(*n* = 226)	(*n* = 679)	(*n* = 227)	(*n* = 226)	(*n* = 226)
All snacks	48.8 ± 0.7%	27.8 ± 0.7%	48.9 ± 0.3%	69.8 ± 0.7%	275 kcal	129 kcal	275 kcal	510 kcal
					(163–436)	(85–163)	(238–317)	(436–632)
Healthy snacks	29.3 ± 0.7%	8.8 ± 0.4%	28.5 ± 0.4%	50.6 ± 0.7%	166 kcal	45 kcal	167 kcal	355 kcal
					(74–286)	(8–74)	(127–198)	(286–461)
Unhealthy snacks	19.5 ± 0.7%	3.2 ± 0.2%	16.4 ± 0.3%	39.2 ± 1.0%	88 kcal	13 kcal	88 kcal	213 kcal
					(30–168)	(0–31)	(67–109)	(168–282)

^1^ % total energy intakes presented as mean ± robust standard error; ^2^ kcal from snacks presented as median (interquartile range); medians were calculated for each definition parameter and each tertile, resulting in sub-group medians that do not sum to the median for the total sample.

**Table 4 nutrients-11-02962-t004:** Associations between snack consumption and dietary outcomes and BMIZ, by snack nutritional quality and consumption metric ^1,2,3,4^.

		Total Energy Intake ^5^	MPA	BMIZ
Model 1:	All snacks consumed between meals + Metric: %TEI	0.10 (−0.002, 0.21)	0.04 (0.005, 0.08) *	−0.15 (−0.34, 0.02)
Model 2:	All snacks consumed between meals + Metric: kcal	0.96 (0.88, 1.04) ***	0.32 (0.29, 0.35) ***	0.03 (−0.15, 0.22)
Model 3:	Healthy snacks consumed between meals + Metric: %TEI	0.21 (0.10, 0.31) ***	0.15 (0.11, 0.18) ***	−0.14 (−0.33, 0.05)
Model 4:	Healthy snacks consumed between meals + Metric: kcal	0.84 (0.75, 0.93) ***	0.33 (0.30, 0.36) ^***^	0.09(−0.28, 0.11)
Model 5:	Unhealthy snacks consumed between meals + Metric: %TEI	−0.06 (−0.17, 0.05)	−0.09 (−0.13, −0.05) ^***^	0.06 (−0.13, 0.25)
Model 6:	Unhealthy snacks consumed between meals + Metric: kcal	0.50 (0.40, 0.60) ***	0.08 (0.04, 0.12) ***	0.14 (−0.05, 0.33)

^1^ Results presented as β (95% CI) comparing highest consumption tertile to lowest; ^2^ Random-effects linear regression with cluster adjustment; model covariates: child age and sex and caregiver wealth status, educational attainment, and caste; ^3^ * *p* < 0.05, ** *p* <0.01, *** *p* < 0.001; ^4^ Snacks were defined as foods consumed outside mealtime and nutrient profiling was conducted to categorize these snacks as unhealthy or healthy. Two measurements of consumption were used: 1) total energy (kcal) from snacks and (2) contribution to total energy intake (%TEI) from snacks. ^5^ Dependent variable log-transformed; results can be interpreted as percent difference between low and high tertiles of consumption using: exp(β)-1*100.

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
