# Peer review of "Exploratory Analysis of Nutritional Quality and Metrics of Snack Consumption among Nepali Children during the Complementary Feeding Period"

_nutrients, 2019, doi:10.3390/nu11122962_

Round 1

Reviewer 1 Report

The revised paper is much improved. However a few concerns remain:

Abstract: A brief definition of “healthy” would be welcome in the abstract.

Line 38. Confusion remains about how “processed” or even “ultra-processed” foods affect the effects studies here. Healthy or unhealthy snacks are defined on the basis of nutrient quality, rather than level of processing (rightfully so). Therefore the addition of the prefix “ultra” does not do anything to clarify the statement about why the consumption of such foods may be concerning. Unless the authors want to explain/clarify/justify what they have in mind here, I would suggest avoiding reference to levels of food processing.

Line 57: “and others” rather than “and other”

Lines 96-98. It is unclear what the percentages are referring to. Percentage of children consuming these foods? If so, then the sentence should be modified.

Line 458: “these data”, rather than “this data”.

Author Response

Dear Reviewer 1,

Thank you very much for your review and comments. We have responded to these comments as outlined below and relevant revisions have been made in the resubmitted manuscript document.

Comments from Reviewer 1:

Abstract: A brief definition of “healthy” would be welcome in the abstract.

RESPONSE: This has been added to the abstract.

Line 38. Confusion remains about how “processed” or even “ultra-processed” foods affect the effects studies here. Healthy or unhealthy snacks are defined on the basis of nutrient quality, rather than level of processing (rightfully so). Therefore the addition of the prefix “ultra” does not do anything to clarify the statement about why the consumption of such foods may be concerning. Unless the authors want to explain/clarify/justify what they have in mind here, I would suggest avoiding reference to levels of food processing.

RESPONSE: This has been revised accordingly.

Line 57: “and others” rather than “and other”

RESPONSE: This has been revised accordingly.

Lines 96-98. It is unclear what the percentages are referring to. Percentage of children consuming these foods? If so, then the sentence should be modified.

RESPONSE: This has been revised accordingly.

Line 458: “these data”, rather than “this data”

RESPONSE: This has been revised accordingly.

Reviewer 2 Report

I am happy with the amendments that the authors have made in response to my comments and can now endorse the manuscript for publication.

Author Response

RESPONSE TO REVIEWER 2:

Dear Reviewer 2,

Thank you very much for your review and comments. We have responded to these comments as outlined below and relevant revisions have been made in the resubmitted manuscript document.

Comments from Reviewer 2:

I am happy with the amendments that the authors have made in response to my comments and can now endorse the manuscript for publication.

RESPONSE: Thank you.

This manuscript is a resubmission of an earlier submission. The following is a list of the peer review reports and author responses from that submission.

Round 1

Reviewer 1 Report

This study examines the nutritional contribution of out-of-meal eating events in 679 young children (1-2 yr-olds) in Nepal. Intake data are extracted from 24-h dietary recalls. The outcomes of interest are total energy intake and dietary adequacy (primary outcomes) and BMIz for age (secondary outcome). The authors use two “definitions” of “snacks”: first snacks are defined as any intake that takes place outside of main meals, second snack input is analyzed in a qualitative (healthy versus non-healthy) perspective.  Two metrics (total energy intake and % daily energy intake) are used. Snack intake represents about one third of total energy intake (when including breastmilk input), and one half of complementary food sources. It critically contributes to daily energy intake and nutrient adequacy in this population of generally lean children (20% stunting and less than 1% overweight). Not surprisingly, benefits are clearer when the snacks are considered “healthy” than “non-healthy”.

There are many interesting aspects to this research. Snack intake is examined in a country experiencing nutritional transition. The children appear to be one rare population in the world where overweight/obesity is not prevalent whereas stunting is frequent. This clearly determines the context of the study. Consequently, the examined behaviors and the nutritional findings should be discussed in this highly specific context. Saying that snacks significantly increase energy intake means radically different things in populations affected by obesity or by stunting. The particular context of the study should be addressed both in the introduction and in the discussion.

There are many conceptual and methodological issues with the paper that require attention. First, the authors claim that they use two definitions of “snacks”. It appears, however, that snacks are always defined as: eating events outside of main meals. Then a further analysis is proposed to examine the healthy/non healthy character of the foods ingested at these out-of -main-meals eating events. If the authors want to define snacks according to their healthy/non healthy character, then all daily eating events should be examined in order to determine if they belong to the “meal” or the “snack” category. From the limited information given in the paper, it seems that some “meals” do include “unhealthy” foods. So clear criteria should be given to determine when and why an eating event is considered a “meal” versus a “snack” based on nature of the foods ingested. Actually, in this work, daily eating events are considered to be “snacks” when they are not one of the 3 main daily meals. That is THE definition of snacks. The text acknowledges this (lines 161-162). Then an analysis of the “snacks” can be done in terms of sub-categories (line 163) of “healthy” or “not-healthy” food choices. The text should avoid confusion between these two levels of analysis (rather than two definitions of snacks).

Another problem is the basis for defining a “snack” food/beverage as “healthy” or “not healthy”. The text tells us that this is based on a nutrient profiling model developed in the UK (FSA).  In that case, the criteria for deciding, in a totally dichotomous fashion, that a food/beverage is “healthy” or “unhealthy” should be provided somewhere in the text. This would be important with studies carried out anywhere in the world. It is all the more important in a population of children affected with stunting and low nutritional adequacy. Among the “unhealthy” food choices are “bakery items”, “instant noodles”, and “juice drinks”, all of which may bring a number of important nutrients to the diet, not to mention energy. Of note, in this perspective, “snacks” are not eating events consumed outside of main meals; they are simply foods and beverages, per se. What the study is examining is what happens to nutrition when “unhealthy” foods are consumed outside of main meals

A consequence of this dichotomous definition of foods/beverages (rather than “snacks) as either healthy or unhealthy is that many claimed effects on nutrition adequacy appear circular: if you ingest “healthy foods”, you end up with a healthy diet. Several instances in the text reflex this circularity.

Title: the title is long. Although the Shakespearian reference is enticing, it is misleading since the study does not really address different definitions of “snacks”. One important aspect of the title is that it sets the scene: Nepali children. The introduction and discussion sections should do the same.

Abstract:

Line 23: Saying that the positive effect of snacks on dietary adequacy was greater when food types were healthy is circular, since dietary adequacy and “healthiness” of foods are defined with the same criteria. If I am wrong, please convince me.

Introduction

The introduction should address the specific context of the study: child nutrition in Nepal, an emerging country experiencing nutritional transition. It should be clear that we are not dealing with overfed kids at high risk of obesity. The nutritional context is of paramount importance. It is also what makes the results novel.

Line 35: do you mean that “processed” foods per se are concerning, or that high-sugar, high-fat foods are concerning? Do you mean that the only way to bring high-sugar, high-fat foods to Nepali children is via processed foods? This could be an interesting idea. Please let us know what you mean here. The role of processed foods can be different in developed versus emerging countries, as suggested recently by Monteiro et al (Cell Metabolism, 2019, 30). This should be addressed somewhere in the paper.

Lines 78 – 80: the snack “definitions” are not really of the same order. One is based on identifying main meals and defining all other eating events as “snacks”. The other appears to be about foods rather than eating events and derive from the former definition: any food/beverage consumed outside of main meals is examined for its “healthy” –“unhealthy” merits. The former seems to characterize eating events while the latter characterizes foods/beverages. In both cases, however, a “snack” is defined as out-of-main-meal intake.

METHODS

Lines 113-114: results are based on one 24-h dietary recall.   Repeated measures of intake were collected on 10% of the sample to account for intra-individual variation. Intra-individual variation should be addressed in the discussion.

Lines 129 and following: The text claims that one definition was “temporal”. Not really. The first definition is a dichotomous categorization of eating events between “meals” and “snacks”, regardless of the time of day when they take place. The present work did not really address the temporal distribution of such events and its potential effects. Temporal analyses of daily eating events do exist (for example Garaulet & Gomez-Abelan, Physiol Behav 2014,134:44-50) and it is important to avoid confusion.

Lines 144 and following: The text suggests that there are 3 main meals in the daily diet of Nepali children (and only 2 for adults). Eating at other moments is considered “snacking”. This seems to be THE definition of “snacks” in this work. The text does not inform about the composition of the various meals, in particular the third “meal” consumed by children in the afternoon. If definitions were based on “healthy” versus “unhealthy”, then it would be important to know the composition of all eating events. The reason why the third “meal” was considered a “meal” rather than a “snack” is unclear.

Lines 153 and following: this second notion is about the quality of foods/beverages consumed outside of main meals. It is based on the UK FSA profiling model. The text should provide the criteria for categorizing a food/beverage as “unhealthy” or “healthy”. When “snacks” (eating events outside of main meals) are composed of more than one food/beverage, do you look at the “healthy-unhealthy” dimension for the whole snack? How? If you don’t, then you are looking at individual foods/beverages rather than “snacks” as eating events. This is confusing.

Lines 166-169: Exploring whether the dietary quality of snack foods is associated with the dietary quality of the diet appears circular.

Line 173: why should the authors worry about “overnutrition” in this population? It would seem more appropriate to examine predictors of undernutrition or stunting.

Lines 185 and following: an average amount (293 g/d) of breastmilk is added to the diet of each child. Is there any way to assess individual differences? If individual differences are likely to be present (as in any biological parameter), then their potential impact of the nutritional outcomes should be discussed.

Lines 223-225: A few problems with this sentence. First saying that food type (healthy or not) could be associated with nutrient adequacy seems circular. Secondly, “these foods” (line 224) seems to refer only to “unhealthy” foods that might be defined as such because they are energy-dense. Beyond the logical issue, an important problem here is the nutritional context: are energy-dense foods “unhealthy” in a context of prevalent undernutrition and stunting?

Table 1 is difficult to read, at least as it appears on the printed page. Beyond the problem of identifying what belongs in the successive rows, I do not understand how parameter combinations 3 and 5 that have to do with healthy versus unhealthy snack contribution to %TEI are both hypothesized to have a negative association with MPA. Do I read this correctly? Similar problem with parameter combinations 4 and 6: healthy and unhealthy snack foods are expected to have the same effects on TEI, MPA and BMIz.

RESULTS

Lines 234-235: Very important observation that should guide the interpretation of the results.

Table 2: The total energy intake is 595 cal. It should be made clear that this excludes energy from breast milk.

Lines 241- 248: very interesting list of “healthy” versus “unhealthy” foods. It is not easy to understand why bakery items and instant noodles should be considered unhealthy in a context of prevalent stunting, low length-for-age and low average BMI.

Lines 258-261: there seems to be an inversion here. The data show greater effect size for consumers with the greatest amount of kcal (0.32) as compared to those with the greatest %TEI (0.04).

Lines 265 and following: showing better nutritional adequacy in consumers of “healthy” snack foods is circular.

Lines 274-275: again a very interesting point that is not addressed as it should be. Snacks, healthy or unhealthy, did not affect BMIz. They contribute to about one third of daily energy intake but they do not affect stunting (20% of the sample), nor overweight (less than 1% of the sample)!

DISCUSSION

The discussion section should address strengths and limitations of the study.

The notion that consuming healthy foods ends up in a healthy diet is circular. A more interesting view of the contribution of “healthy” foods consumed as snacks in children’s diet could be presented in the context of nutritional transition. While children obviously required enough energy to avoid stunting, a consideration of the nutrient density of their snack foods (and main meal foods) is highly important. The notion that snacks can exert a positive or negative impact on the diet is not novel either. A discussion of various snack definitions in the literature and the ambivalent contribution of snacks in the daily diet can be found in Bellisle, Physiol Behav 2014; 134:38-43.

One dimension that is not addressed at all in the paper and could be of importance is the cost of the various snack foods. Monteiro et al (Cell Metabolism 2019; 30) stress that traditional foods are less expensive than processed foods in emerging countries, which could determine choice and frequency of intake. The economic background of healthy/unhealthy snack food choices should be discussed.

CONLUSIONS

This study did not address “when a snack is consumed” (line 396). Eating events here were defined as either main meals or snacks, regardless of the time of consumption, which is another aspect of intake (Garaulet & Gomez-Abellan, Physiol Behav, 2014; 134:44-50). Please amend.

The observation that consuming healthy snack foods contributes to a healthy diet is circular. Beyond this, insisting that recommendations should encourage the selection of healthy foods/beverages and discourage unhealthy ones (lines 401-402) is superfluous. The authors should improve the interest and novelty of their conclusions by (among other things) discussing their findings in the very special context of nutritional transition.

There are a few typos here and there in the text. For example, line 211 “tertiles”; Line 56 “J.A.”

Reviewer 2 Report

This manuscript reports on the consumption of snack food in infants aged 12 to 23 months in Nepal. It provides a useful account of the topic and does a good job of explaining why the topic is so difficult to analyse. Please see my small number of notes on the manuscript itself.

Thank you for allowing me to review this manuscript.
